# Cognitive Rehabilitation Improves Functional Vision Skills in Children with Cerebral Visual Impairment

**DOI:** 10.3390/brainsci15060590

**Published:** 2025-05-29

**Authors:** Zeynep Celik Turan, Esra Aki

**Affiliations:** 1Occupational Therapy Division, Brunel University of London, Uxbridge UB8 3PH, UK; 2Occupational Therapy Department, Faculty of Health Sciences, Hacettepe University, Ankara 06100, Türkiye; esraaki@hacettepe.edu.tr

**Keywords:** vision disorders, therapy, activities of daily living, participation, cognition

## Abstract

**Background/Objectives**: Cerebral visual impairment (CVI) is the leading cause of visual disability in children, resulting from damage to the brain’s visual processing pathways. Although ocular structures may be intact, functional vision, the use of vision in daily tasks, is often significantly affected. Cognitive Rehabilitation Therapy (CRT) has shown potential to enhance cognitive functions through neuroplasticity, yet its influence on functional vision remains underexplored. This exploratory pilot study aimed to examine whether CRT could improve functional vision in children with CVI by targeting underlying cognitive deficits. **Methods**: A single-arm pre–post intervention study was conducted with nine children aged 7–11 years diagnosed with CVI. Participants received 20 individualized CRT sessions over ten weeks, integrating principles from the Model of Visual Functioning. Functional vision was evaluated using the Gazi Functional Vision Assessment Instrument (GFVAI), while cognitive improvements were measured with the Dynamic Occupational Therapy Cognitive Assessment for Children (DOTCA-Ch) and the Motor-Free Visual Perception Test-4 (MVPT-4). Data were analyzed using Wilcoxon signed-rank tests. **Results**: Statistically significant improvements were observed in GFVAI domains such as light sensitivity, distant visual field, focusing, maintaining focus, and obstacle avoidance (*p* < 0.05 to *p* < 0.01). Qualitative analysis also indicated a shift from low/moderate to good/very good performance in most functional vision categories. Cognitive domains, including visual-motor organization, thinking operations, and spatial perception, showed significant gains. **Conclusions**: CRT may support improvements in functional vision by enhancing cognitive processes in children with CVI. This therapist-led approach is feasible, adaptable, and holds promise for widespread application in pediatric rehabilitation.

## 1. Introduction

Cerebral visual impairment (CVI), also referred to as cortical visual impairment, is a form of vision loss resulting from damage to the visual cortex, the posterior visual pathways, or both. Although ocular structures may remain intact, the brain’s ability to process and interpret visual information is disrupted, leading to significant visual deficits. This may include low vision, which is commonly defined as best corrected visual acuity between 20/60 and 20/200 in the better eye or a visual field below 20 degrees. In many functional classifications, this level of vision corresponds to a 10–30% impairment and may significantly impact everyday activities such as reading, navigation, and self-care, even when ocular health is preserved [1,2].

CVI may result from a variety of neurological conditions, including hypoxic-ischemic encephalopathy, epilepsy, focal brain lesions, central nervous system infections, traumatic brain injury, and cerebral palsy [1,3,4,5]. It is estimated that over 30–50% of the brain is involved in visual processing, encompassing not only the primary visual cortex (occipital lobe) but also associated areas in the parietal, temporal, and frontal lobes responsible for visual attention, spatial awareness, motion detection, and object recognition. This distributed visual network explains why damage in many different regions of the brain can lead to visual difficulties, even when the eyes are structurally normal [1,3,6]. While CVI is a lifelong condition, it is increasingly recognized as a developmental disorder that may evolve with targeted intervention, particularly through neuroplastic processes [6].

Functional vision refers to the individual’s ability to use their visual capacity in meaningful, everyday activities such as reading, navigating environments, engaging in household tasks, or watching television. Significant vision loss due to CVI can severely limit participation in daily life, especially when remaining visual function is insufficient [7,8,9]. The effective use of vision depends on the integration of several interrelated factors. According to the Model of Visual Functioning (MVF) developed by Corn [10], functional vision comprises three main components: (1) stored and usable individual characteristics (e.g., cognitive abilities such as experience, memory, concept development, communication, problem solving, sensory integration, psychological profile such as attention, and perception such as part/whole, figure/ground, closure, sequence), (2) environmental cues (e.g., contrast, lighting, spatial layout, color, and timing), and (3) visual skills (e.g., acuity, light and color perception, brain functions, visual field, oculomotor control, and visual perception). Improvements in any of the three MVF components can dynamically enhance functional vision by influencing how visual information is acquired and used in context [8]. These three components reflect how vision is not an isolated sensory process but an integrated function involving higher-order brain systems such as attention, memory, and executive processing, all of which reflect the cognition’s active role in vision. Recent guidelines for CVI rehabilitation emphasize the importance of structuring interventions around cognitive support and environmental adaptations, consistent with the MVF framework [11].

Research further suggests that functional vision in children with CVI may be improved by leveraging the brain’s capacity for neuroplasticity [12,13]. Neuroplasticity refers to the brain’s ability to reorganize and form new neural connections in response to experience and targeted stimulation. While this process occurs naturally throughout development, it can also be enhanced through structured interventions such as cognitive rehabilitation [14,15,16]. CRT is a therapeutic approach aimed at restoring or compensating for impaired cognitive skills, such as attention, memory, visual processing, information processing, and executive functions, following brain injury or dysfunction. CRT typically involves a structured process that includes identifying cognitive strengths and weaknesses, training underlying cognitive capacities, teaching compensatory strategies, and applying these strategies in real-life functional activities [17,18].

CRT emphasizes the individualization of intervention based on the client’s learning style, cognitive profile, and functional needs and can be delivered in both clinical and naturalistic environments. Aligning intervention activities to a child’s cognitive developmental level has been shown to optimize visual outcomes such as visual attention, tracking, and spatial awareness, which contribute directly to the development and use of functional vision [19]. The goal is to enhance participation in everyday activities by improving the cognitive processes that support them. In line with this, Bennett et al. [20] highlight the benefits of applying multisensory, cognitively enriched strategies in naturalistic contexts to support functional vision. Previous studies suggest that cognitive training can improve specific visual functions in children with CVI by activating neuroplastic mechanisms. While full restoration of functional vision is not typically reported, gains in areas such as visual field use, attention, and visual-motor coordination have been documented following interventions targeting cognitive and attentional processes [13,21,22].

In light of this theoretical framework, it may be hypothesized that improving cognitive skills can lead to measurable improvements in functional vision, particularly those underlying attention, visual perception, and executive function. Since cognition is one of the foundational components of the MVF, strengthening it through CRT has the potential to improve functional vision. While prior studies have examined neuroplasticity and cognitive performance in children with CVI [13,21,22], to our knowledge, no research has directly evaluated the impact of CRT on functional vision outcomes in this population. This preliminary exploratory study aimed to investigate the effect of cognitive rehabilitation training on functional vision skills in children with low vision due to CVI.

## 2. Materials and Methods

This study employed a single-arm pre- and post-test design and was conducted in accordance with the Declaration of Helsinki, with ethical approval obtained from the *** University Ethics Committee (Approval No: 20181022). Informed consent was obtained from both the participating children and their legal guardians. All participants were informed about the purpose, procedures, potential risks and benefits of the study, and their right to withdraw at any time without consequence. Following the collection of signed consent forms from both the children and their guardians, the study protocol was initiated. Confidentiality and anonymity were ensured throughout the process, with each participant being given a code based on their initials and enrollment numbers (EA1, ZCT2) during the initial recruitment process. This code was used in the records of other assessment tools and case report forms. Sociodemographic data were collected, and standardized assessments were administered prior to the intervention. An individualized training program was then developed for each child. After completing 20 sessions of cognitive rehabilitation training, the same standardized assessments were re-administered to evaluate post-intervention outcomes. All assessments were completed by the first author, an experienced occupational therapist, during one-to-one sessions with each child.

### 2.1. Participants

Participants were recruited through referrals made by both private and public sector ophthalmologists to the Low Vision Rehabilitation Unit of the *** University Department of Occupational Therapy. The children were eligible for inclusion if they met the following criteria: (a) aged between 6 and 11 years, an age range selected to reflect a developmental window in which visual skills are typically consolidated (by age 6) and cognitive abilities, such as abstract thinking and spatial perception, are emerging (from around age 11) [23]; (b) had visual acuity ranging from 20/60 to 20/200 based on Snellen scores, measured with their current prescription glasses or optical aids if used, as confirmed by an ophthalmologist [2]; (c) demonstrated the ability to cooperate with standardized assessment procedures during the initial evaluation phase (e.g., able to complete MVPT tasks without full withdrawal or visual inaccessibility); and (d) voluntarily agreed to participate in the study. Children with a diagnosis of severe cognitive and/or motor impairments (e.g., cerebral palsy with significant motor disability, mild intellectual disability recorded in their current medical history), as well as those who could not attend the training sessions two times per week, were excluded. The study initially aimed to recruit 20 children. Nineteen children were referred to the program by public and private ophthalmologists. Of these, 10 children met the inclusion criteria and were enrolled in the CRT intervention. One child was withdrawn due to a change in family circumstances, resulting in a final sample of nine children who completed the 10-week training and were included in the analysis.

### 2.2. Tools

#### 2.2.1. Sociodemographic and Background Form

Sociodemographic information about age, gender, primary diagnosis and medical history, vision rate, onset of visual impairment, family history, medical history, school, grade, type of class (special education class or mainstream), optical aids and medicines used were recorded.

#### 2.2.2. Gazi Functional Vision Assessment Instrument (GFVAI)

The GFVAI, developed by Şafak et al. [24,25], is a standardized tool designed to evaluate the functional vision of students with low vision in educational settings. The GFVAI was specifically created to assess how children use their residual vision in daily life. The tool demonstrated high internal consistency with a mean Cronbach’s alpha (α) of 0.92, indicating excellent reliability across its subcomponents.

In addition to Cronbach’s alpha, its psychometric robustness is supported by confirmatory factor analysis, which validated the theoretical two-factor model (near and distance vision skills), and by discriminant analysis, which confirmed the tool’s ability to accurately classify students into functional vision levels (weak, moderate, good). Furthermore, inter-rater reliability was ensured using both the Many-Facet Rasch Model and generalizability theory, with G coefficients of 0.99 and Phi coefficients of 0.98, reflecting excellent reliability across raters and conditions.

The GFVAI comprises two main dimensions: Near Vision Skills and Distance Vision Skills. The two main dimensions are further divided into 14 sub-dimensions:Near Vision Skills (8 subdimensions): Focusing, Maintaining Focus (on single and two objects), Monitoring/Scanning, Color Vision, Light Sensitivity, Image Recognition, Near Visual Field, and Use of Writing Tools.Distance Vision Skills (6 subdimensions): Distance Visual Field (central, left, right), Distance Reading and Viewing, Object/Person Recognition, Avoiding Obstacles, Avoiding People, and Navigating Stairs.

Each subdimension is scored individually based on structured performance tasks. Scores are categorized into four qualitative levels: none, weak, moderate, or good. “None” indicates that the child cannot complete the visual task even with support, while “Good” reflects consistent and independent visual performance. Intermediate categories capture partial or inconsistent use of vision. For example, in the subdimension “Avoiding Obstacles”, a child who consistently bumps into objects in an open space would be rated as “None”, while a child who navigates around obstacles independently and smoothly would receive a “Good” rating. Intermediate levels (“Weak” and “Moderate”) reflect partial success, support needs, or inconsistencies. Rather than producing a cumulative total score, the instrument emphasizes profile-based interpretation, allowing both quantitative analysis (numerical scores per subskill) and qualitative evaluation (functional description of ability). Each subdimension of the GFVAI is scored based on structured performance tasks, with specific point allocations reflecting the child’s ability to perform visual tasks at varying distances and conditions. For instance, in the “Maintaining Focus on a Single Object” subdimension, the child is assessed on their ability to maintain focus on a figure presented at distances of 60 cm, 40 cm, and 20 cm. Points are awarded based on the distance at which the child successfully maintains focus, with higher points for longer distances. The total score determines the performance category: 0–13 points indicate ‘Weak’ performance, 14–24 points ‘Moderate,’ and 25–32 points ‘Good.’ This scoring approach allows for a nuanced understanding of the child’s functional vision capabilities.

This detailed assessment method supports individualized educational planning and intervention design, making the GFVAI a highly applicable tool for occupational therapists, educators, and vision specialists working with children with visual impairments. Further information on the development of the test can be found in [25], and full scoring guidelines can be referred to in [24].

#### 2.2.3. Dynamic Occupational Therapy Cognitive Assessment for Children (DOTCA-Ch)

The DOTCA-Ch is a standardized, performance-based tool designed to evaluate cognitive functioning and learning potential in children aged 6 to 12 years. Developed within a dynamic assessment framework, the DOTCA-Ch measures cognitive modifiability through both baseline performance and response to structured mediation. It includes 22 subtests across five cognitive domains (scores): Orientation (1–16), Spatial Perception (1–12), Praxis (1–44), Visuomotor Organization (7–35), and Thinking Operations (7–35). Each subtest is scored independently, allowing for a detailed cognitive profile, according to the DOTCA-Ch manual. Higher scores indicate better performance. The tool demonstrates excellent reliability, with Cronbach’s alpha values ranging from 0.87 to 0.99, supporting its consistency across domains and applications [26].

#### 2.2.4. Motor-Free Visual Perception Test-Fourth Edition (MVPT-4)

The MVPT-4 is a standardized assessment tool designed to measure visual perceptual abilities independent of motor involvement [27]. Suitable for individuals aged 4 to 80+ years, the MVPT-4 is particularly beneficial for evaluating children with motor impairments or neurological conditions, such as cerebral visual impairment (CVI). The tool comprises 45 items assessing five core domains: Visual Discrimination, Form Constancy, Visual Memory, Visual Closure, and Spatial Relationships. Responses are given verbally or by pointing, eliminating the need for fine motor coordination, as described in the MVPT-4 administration guide [27]. The MVPT-4 produces a single total raw score ranging from 0 to 45, with one point awarded for each correct item. The test has demonstrated strong psychometric properties, with internal consistency coefficients typically exceeding α = 0.80, supporting its reliability across age groups [27].

### 2.3. Training Protocol

In developing the protocol, CRT was integrated with the MVF [8,10] as the foundational framework. The intervention program was created individually for each child based on their cognitive profile, age, visual perceptual skills, and level of functional vision, in alignment with CRT’s structured process and the MVF’s three interrelated components.

Training was conducted over 20 sessions (twice weekly, 60 min each) in the *** University Low Vision Rehabilitation Unit. The protocol followed CRT’s four-step approach: (1) informing the child and family about the intervention goals and expected outcomes, such as improved attention and memory for visual and auditory input; (2) strengthening underlying cognitive functions through age-appropriate, vision-adapted tasks drawn from an evidence-based activity pool; (3) supporting the development of internal and external compensatory strategies through structured games and collaborative reflection; and (4) applying cognitive strategies to functional tasks in real-life environments such as school and home, with the active involvement of caregivers and teachers.

Activities were individually tailored to target attention, visual processing, memory, information processing, and executive functions. This individualization followed a structured clinical decision-making process informed by baseline assessments (GFVAI, DOTCA-Ch, MVPT-4), each child’s progress through CRT stages, and the MVF framework. For example, while earlier sessions placed greater emphasis on foundational skills such as attention and visual processing, later sessions progressively shifted focus toward memory, information processing, and executive functioning. Even in the final session, activities often began with brief attention tasks but prioritized higher-level cognitive strategies. This progression reflects the hierarchical nature of cognitive skill development emphasized in CRT and aligns with the literature highlighting the benefits of stage-based adaptation in pediatric cognitive rehabilitation. The MVF framework informed modifications related to individual characteristics (e.g., sensory-motor abilities, motivation, emotional readiness), environmental clues (e.g., lighting, contrast, spatial layout), and visual skills (e.g., acuity, field use, oculomotor control). Challenges were introduced as appropriate, respecting each child’s motivation, endurance, and functional vision profile. Further methodological details, including specific activity examples and CRT stage alignment, are provided in Appendix A.

To ensure fidelity of intervention delivery, all sessions were conducted by the same experienced therapist using a standardized CRT protocol. A session log was maintained to document the activities completed, any adaptations made, and the child’s response. This approach ensured consistency across participants while allowing for individualized application of the intervention principles.

### 2.4. Analysis

The primary outcome was improvement in functional vision, assessed using the GFVAI. Secondary outcomes included changes in cognitive performance, measured by the DOTCA-Ch, and visual perceptual skills, evaluated using the MVPT-4. Outcome assessments were conducted by the same therapist who delivered the intervention; therefore, blinding was not applied and should be considered a methodological limitation.

Descriptive statistics were used to summarize the study data. For numerical variables, minimum, maximum, mean, and standard deviation values were reported. Frequencies were used for categorical variables. Given the small sample size (*n* = 9) and the non-parametric distribution of the data, the Wilcoxon signed-rank test was employed to compare pre- and post-intervention scores for the numeric results of all outcome measures.

All statistical analyses were conducted using IBM [28] SPSS Statistics Version 23.0. A post hoc power analysis and effect size calculation were performed using G*Power 3.1.9.2 [29] to assess the statistical power of the observed effects. Graphs were generated using ChatGPT-4 [30] based on data tables exported from SPSS. All visualizations were cross-checked against statistical outputs to ensure accuracy, and final versions were reviewed and approved by the authors.

## 3. Results

A total of 19 children and their families were referred by ophthalmologists to the study if they had low vision and were interested in taking part in this study. Of these, ten were excluded: eight children were unable to cooperate with standardized assessment procedures, one child did not attend the sessions regularly, and one child was withdrawn as the parents decided to prioritize academic tutoring instead of this study. The final sample consisted of nine children who completed the intervention and both pre- and post-assessments.

The median age of participants was 9 years, with a range of 7 to 11 years. The sample included four girls and five boys. All children had cerebral visual impairment caused by natal factors. Regarding the use of optical aids, four children used none, four used refractive eyeglasses, and one child used telescopic glasses. One participant had a history of spasticity and had undergone surgical intervention. Six children were using anti-epileptic medications, and only one child had a family history of visual impairment. Educational placements varied: four children attended inclusive mainstream classrooms, two were enrolled in special education classes within mainstream schools, and three attended schools for the visually impaired. At the time of the study, the children were in the second through sixth grades. Table 1 indicates the individual characteristics of each participant.

### 3.1. Quantitative Results

The results of the pre- and post-intervention comparisons are presented in Table 2, including means, standard deviations (SD), Wilcoxon signed-rank test statistics (z), and *p*-values. Statistically significant improvements (* *p* < 0.05, ** *p* < 0.01) were observed across several domains of functional vision and cognitive functioning, indicated by negative z values.

Within the GFVAI, children demonstrated significant gains in both near and distant visual skills. For near vision, improvements were found in focusing, maintaining focus, tracking, near visual field, and color vision (*p* < 0.05), while light sensitivity and image recognition yielded highly significant differences (*p* < 0.01). Regarding distant vision, statistically significant improvements were observed in distance reading/viewing, obstacle avoidance, and navigating around people (*p* < 0.05). Highly significant differences were found in the distant visual field and stair navigation (*p* < 0.01). However, changes in writing tool use, voucher reading, and object/person recognition did not reach statistical significance.

In the cognitive outcomes, as assessed by the DOTCA-Ch, significant improvements were found in orientation and spatial perception (*p* < 0.05). More robust changes were observed in praxis, visual motor organization, and thinking operations (*p* < 0.01). Additionally, the MVPT-4 results indicated a statistically significant improvement in visual perceptual skills (*p* < 0.01).

### 3.2. Qualitative Results

Qualitative data from the GFVAI are presented in Figure 1, which reflects group-level distributions. Prior to the intervention, most children demonstrated low to moderate ratings in functional vision skills; no child had a “none” rating in any of the categories. Following the training program, the trend shifted clearly toward moderate to good performance across both near and distant visual tasks.

The most notable qualitative gains were observed in focusing, maintaining focus, near visual field, light sensitivity, and distant visual field. In many cases, children moved from a rating of “weak” before the training to “good” afterwards. Observable functional improvements were also recorded in skills such as obstacle avoidance, walking without collisions, and navigating stairs with increased independence.

### 3.3. Effect Size and Power Analysis

A post hoc power analysis was conducted based on the results of the Wilcoxon signed-rank test comparing the means of two dependent groups. The estimated effect size was approximately 1.5, which indicates a very large effect. Given an alpha level of 0.05 and a total sample size of nine participants, the statistical power of the study was calculated as 0.99, suggesting that the sample was sufficiently powered to detect the observed effects.

## 4. Discussion

This exploratory study aimed to investigate whether improving cognitive skills through a structured cognitive rehabilitation protocol could enhance functional vision in children with cerebral visual impairment (CVI). While the findings demonstrate statistically significant improvements, they should be interpreted as preliminary, given the study’s design and scope. The findings support the hypothesis, demonstrating significant improvements in both near and distant visual skills, as well as cognitive domains such as orientation, spatial perception, praxis, visual-motor organization, thinking operations, and visual perception. Given the exploratory nature of this study, the findings should be interpreted as preliminary and hypothesis-generating, requiring validation in future controlled research.

The findings of this study contribute to the growing body of evidence suggesting that functional vision is shaped not only by ocular integrity but also by higher-order cognitive and perceptual processes [13,20,21,22,31]. The Model of Visual Functioning (MVF) clearly positions cognition as one of the three foundational components of functional vision [8,10], alongside visual skills and environmental cues. The observed improvements in both cognitive and functional vision domains in this study underscore the interdependence of these systems. Functional vision goes beyond basic visual abilities such as acuity or field integrity, relying heavily on cognitive mechanisms including attention, executive functioning, and spatial reasoning. In our intervention, it is likely that gains in cognitive capacity supported more effective use of visual information—and vice versa. While the precise direction of influence cannot be determined without neurophysiological data, the co-occurring improvements in both domains may reflect a shared underlying mechanism, such as neuroplastic adaptation. Rather than treating the overlap between cognition and vision as a limitation, this interdependence highlights the potential for CRT to serve as a dual-benefit intervention in pediatric rehabilitation.

The recent literature supports integrating cognitive and environmental strategies in CVI rehabilitation. For instance, Bennett et al. [20] advocate for real-life, multisensory, and cognitive-based approaches to support functional vision—principles that align with the therapist-led, everyday-contextual nature of our intervention. However, unlike their general framework, our study operationalized these strategies within a structured cognitive rehabilitation protocol specifically grounded in the Model of Visual Functioning (MVF). Similarly, Weden et al. [19] emphasized the importance of tailoring interventions to cognitive developmental age. While we adopted this principle during activity adaptation, our study extended it by systematically applying it within CRT stages targeting multiple cognitive domains. Fonteyn-Vinke et al. [11] have also highlighted the need for cognitively supportive rehabilitation, yet their work focused primarily on guideline development. In contrast, our study presents a practical, therapist-implemented intervention protocol with measurable outcomes, demonstrating how such guidelines can be translated into functional, real-world rehabilitation.

While previous research often focused on isolated cognitive domains such as attention [22] or compensation [32], the current study targeted multiple cognitive areas simultaneously. This broader scope reflects the complexity of real-world visual functioning and was guided by the MVF model. Targeting multiple domains such as attention, memory, visuomotor coordination, and executive functioning may increase the ecological validity of the intervention and promote generalization to diverse daily tasks. However, it also introduces complexity in evaluating which cognitive processes contributed most directly to the observed improvements. Moreover, unlike earlier works that overlook environmental and cognitive adaptation, our protocol aligned each activity with MVF components and real-life contexts. Conducting the intervention in natural settings, such as school and home, increases relevance and transferability but also introduces variability due to uncontrolled environmental factors. This is especially important to acknowledge in a small-sample exploratory study, where individual differences and contextual variability may influence outcomes. One notable example is the study by Malkowicz, Myers, and Leisman [13], which reviewed early CRT efforts in CVI and emphasized visual-motor and attention-based training but primarily within clinical environments and often focused on specific modalities. While the previous work demonstrated that some visual gains could occur through cognitive stimulation, it did not integrate environmental adaptation or a multi-domain cognitive strategy within a functional vision model. Our study builds upon and expands this perspective by embedding CRT within an MVF framework, combining cognitive rehabilitation with real-life functional and meaningful visual tasks and tailoring activities to both environmental and personal factors—a therapeutic approach more aligned with current neurodevelopmental rehabilitation principles [31,33,34].

The present study aligns with existing theories about the potential role of neuroplasticity in visual function recovery. Williams [6] emphasizes that although CVI is a lifelong condition, it is not unchangeable; structured cognitive engagement can facilitate developmental gains through neuroplastic mechanisms. While our study did not directly measure changes in brain structure or function, the observed improvements in cognitive and functional vision skills are consistent with the idea that targeted cognitive stimulation may facilitate adaptive neural responses. These findings echo previous work by Korsunskaya et al. [35], who reported functional and emotional improvements in neurologically affected children in association with neuroplastic processes. Thus, while causality cannot be confirmed, our results are suggestive of potential neuroplastic benefits that warrant further investigation through neurophysiological or imaging studies. The findings are further reinforced by Waddington and Ellis [31], suggesting that developmental plasticity may extend beyond traditionally assumed sensitive periods.

In addition to potential neural benefits, the intervention used in this study offers several practical advantages that strengthen its clinical relevance. Functional vision gains like those observed here are associated with better quality of life outcomes [36], emphasizing the broader significance of this research. Importantly, the intervention used here required no laboratory setting, high-tech equipment, or invasive techniques. It was delivered using simple, adaptable materials in natural settings by a trained occupational therapist over 20 structured sessions, in addition to pre- and post-intervention assessments. This format reflects a moderate time commitment, comparable to standard cognitive or low vision rehabilitation programs, and remains feasible for integration into outpatient or school-based service delivery without requiring extensive personnel resources. This makes the protocol highly scalable, particularly for low-resource contexts, and strengthens the case for widespread implementation in occupational therapy. In addition to its feasibility, this approach allows therapists to address visual-cognitive integration through structured, developmentally aligned activities—something not commonly emphasized in traditional vision-focused interventions. As such, CRT offers a promising adjunct to conventional methods by targeting underlying cognitive mechanisms that support the functional use of vision.

In our study, not all functional vision domains showed statistically significant improvement. Specifically, writing, reading receipts, and identifying people did not demonstrate measurable gains. These skills often rely heavily on visual acuity and fine visual-motor coordination, areas that may not be fully targeted through cognitive rehabilitation alone. In the case of reading receipts, success often depends not only on visual perception but also on literacy-specific recognition and contextual understanding, which were not directly addressed in our training. Similarly, identifying people requires higher-level social-perceptual processes such as face recognition, emotion detection, and interpersonal memory—cognitive domains related to communication and social cognition, which were not a primary focus of the CRT protocol used in this study. Çakmak et al. [37] have observed that children with residual vision in mainstream schools tend to show higher functional vision capacity, likely due to their exposure to visually stimulating learning environments. While our participants shared these characteristics, the lack of progress in certain school-related skills suggests that visual-cognitive gains may not directly translate into academic or interpersonal visual performance. Grbović and Stanimirov [38] emphasized that visual tasks such as writing often require a coordinated, school-based approach involving teachers, therapists, and families. In parallel, Aki et al. [39] demonstrated that motor skills training can significantly improve visual-motor integration in children with low vision. Therefore, future interventions may benefit from integrating CRT with structured motor training and school-based collaboration to support more complex visual tasks like writing and person identification. It is also possible that skills such as writing require a longer period of visual adaptation and that functional improvements may emerge over a more extended timeline than was captured in this 10-week intervention.

A key limitation is the absence of a control group. While the within-subjects design allowed for measurement of change over time, the lack of a randomized control or comparison group limits the ability to attribute observed improvements solely to the intervention. The decision not to include a control group was influenced by the exploratory nature of the study and ethical considerations regarding withholding potentially beneficial therapy from children with significant visual and cognitive challenges. Consequently, alternative explanations for the observed improvements, including natural developmental progression or environmental exposure, cannot be fully ruled out. For example, over the 10-week summer period, some cognitive or functional gains may have occurred through maturation or exposure to informal learning opportunities such as home-based activities, regular special education services, or extracurricular programs.

In addition to the absence of a control group, several limitations must be acknowledged. While the post hoc power analysis indicated a very large effect size (r = 1.5) and high statistical power (0.99), it is important to interpret these findings cautiously given the small sample size and non-parametric nature of the data. Although such calculations suggest meaningful change, the absence of a control group, the variability of individual responses, and the flexible, individualized nature of the CRT protocol make it difficult to attribute observed improvements to specific intervention components with certainty. Future studies should apply more standardized fidelity checks and control conditions to rigorously evaluate intervention effects across subpopulations. The study’s eligibility criteria may also have introduced selection bias, as participants likely had higher levels of cognitive and behavioral readiness, as well as greater family support. As a result, the findings may not apply to children with more complex profiles or limited access to structured rehabilitation. Future studies should also explore strategies to improve accessibility and participation, such as incorporating observational assessments such as CVI Range [4] or CVI-PIMD [40] to determine readiness for cognitive and visual tasks. These approaches may help identify children who could benefit from tailored adaptations or support, thereby reducing attrition and increasing the inclusivity of CRT interventions. These limitations highlight the need for larger-scale studies that allow more diverse structures. Additionally, all intervention sessions and outcome assessments were conducted by the same therapist, which, while ensuring fidelity, also introduces the potential for assessment bias due to the lack of blinding. These limitations underscore the need for future large-scale, controlled studies with diverse participant profiles and independent outcome evaluation.

Future research should prioritize the implementation of randomized controlled trials (RCTs) to validate the efficacy of CRT protocols, isolate their specific effects, and determine their generalizability across broader populations of children with CVI. RCTs would allow for stronger causal inferences and help differentiate specific intervention effects from spontaneous improvements or placebo effects. Furthermore, comparing CRT to other interventions, such as visual stimulation alone or environmental adaptation training, could help delineate the unique contribution of cognitive components to functional vision outcomes. Despite this limitation, the findings from this exploratory study provide a robust foundation for further hypothesis-driven research using rigorous experimental designs. While the current study focused on improvements in functional vision skills as measured by standardized tools, future research should extend its focus to include the participation of children with CVI in everyday activities and consider how contextual factors such as school type may influence intervention outcomes. Functional vision is only one component of occupational performance; it remains to be understood how the observed improvements influence children’s engagement in school, play, self-care, and social interaction. Moreover, incorporating the perspectives of families, who play a key role in supporting children’s participation, would provide critical insight into the real-life impact and relevance of such interventions. Exploring how families perceive functional gains, challenges in daily routines, and environmental barriers can inform more comprehensive, family centered rehabilitation approaches. These perspectives are essential for designing interventions that not only enhance functional vision but also facilitate meaningful participation and inclusion in daily life. Although the improvements in functional vision and cognitive functioning are encouraging, they cannot be overgeneralized. The findings provide early insights into the potential role of CRT but require confirmation through larger and more rigorous trials.

## 5. Conclusions

In light of our exploratory study, the findings suggest that functional visual skills in children with CVI may be improved through cognitive rehabilitation. Significant gains were observed in both cognitive domains (e.g., visuomotor organization, spatial perception) and functional vision tasks aligned with the MVF model. These results support the role of cognition in vision rehabilitation and highlight the potential of CRT as a useful approach. However, further research with larger samples and controlled designs is needed to confirm efficacy and assess broader impacts on participation and quality of life.

## Figures and Tables

**Figure 1 brainsci-15-00590-f001:**
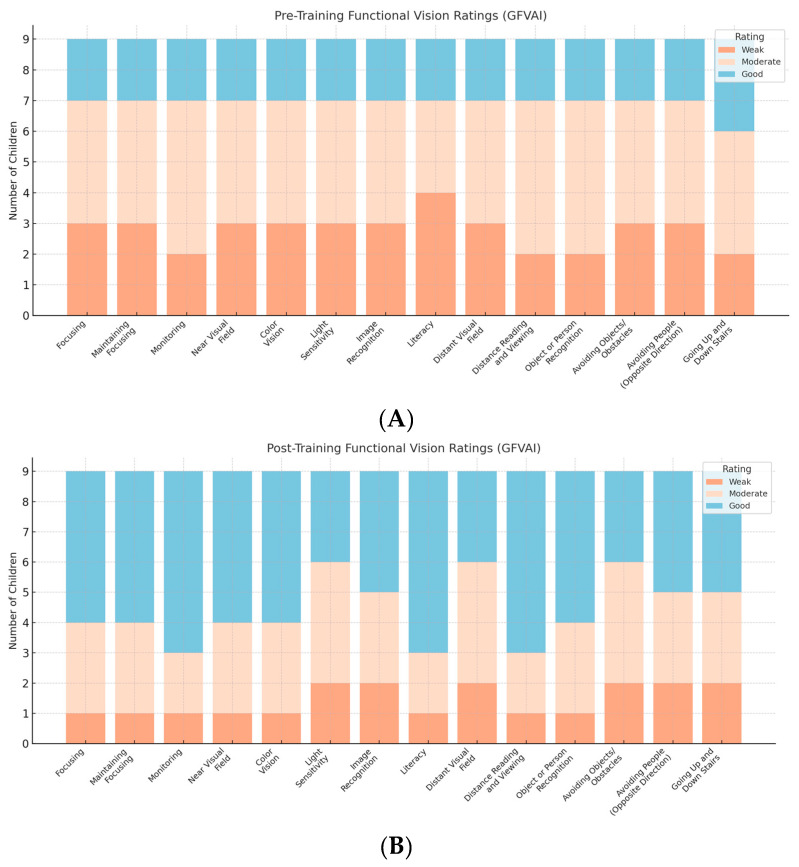
Qualitative data from the GFVAI; (**A**) prior to training; (**B**) after the training.

**Table 1 brainsci-15-00590-t001:** Individual characteristics of participants (*n* = 9).

ID	Age	Gender	Diagnosis	Optical Aid	Medical History	Family History	Medication	Educational Placement	School Grade
P1	7	Male	CVI, Color Blindness, Photophobia	Glasses	No	No	No	Inclusive Mainstream Classroom	2nd
P2	10	Male	CVI	Glasses	No	No	No	Inclusive Mainstream Classroom	3rd
P3	9	Male	CP CVI	Glasses	Spasticity operation	Yes	Yes	Inclusive Mainstream Classroom	4th
P4	11	Male	CVI	None	No	No	No	Inclusive Mainstream Classroom	5th
P5	7	Female	CVI	Glasses	No	No	Yes	Special Ed in Mainstream School	5th
P6	11	Female	CVI	None	No	No	Yes	Special Ed in Mainstream School	5th
P7	11	Male	CP, CVI, Epilepsy Nystagmus	None	No	No	Yes	School for the Visually Impaired	6th
P8	11	Female	CVI, Epilepsy Nystagmus	None	No	No	Yes	School for the Visually Impaired	6th
P9	10	Female	CVI	Telescopic Glasses	No	No	Yes	School for the Visually Impaired	6th

**Table 2 brainsci-15-00590-t002:** Quantitative findings regarding the comparison of the results before and after the intervention (*n* = 9).

Skill	Pre-Training X ± SD	Post-Training X ± SD	z	*p*
**GRVAI-Near Vision Skills**				
Focusing	25.1 ± 8.1	36.0 ± 6.0	−2.410	0.016 *
Maintaining Focusing	19.3 ± 8.1	28.7 ± 4.0	−2.371	0.018 *
Monitoring	36.9 ± 19.2	54.7 ± 11.1	−2.201	0.028 *
Near Visual Field	12.0 ± 3.3	16.1 ± 0.8	−2.555	0.011 *
Color Vision *	42.0 ± 14.0	54.2 ± 10.2	−2.366	0.018 *
Light Sensitivity *	1.0 ± 0.0	3.0 ± 0.0	−3.000	0.003 **
Image recognition	1.8 ± 0.4	2.8 ± 0.4	−2.714	0.007 **
Literacy	2.1 ± 0.8	2.3 ± 0.7	−1.414	0.157
**GRVAI-Distant Vision Skills**				
Distant Visual Field	19.7 ± 6.00	36.4 ± 5.5	−2.670	0.008 **
Distance Reading and Viewing	1.4 ± 0.7	2.1 ± 0.6	−2.449	0.014
Object or Person Recognition	1.8 ± 0.7	2.2 ± 0.7	−1.633	0.102
Avoiding Objects/Obstacles	1.7 ± 0.5	2.4 ± 0.5	−2.333	0.020 *
Avoiding people from the opposite direction	1.6 ± 0.5	2.3 ± 0.5	−2.333	0.020 *
Going Up and Down Stairs	1.8 ± 0.4	2.7 ± 0.7	−2.530	0.011 *
**DOTC-Ch**				
Orientation	5.5 ± 5.2	8.1 ± 4.6	−2.388	0.017 *
Spatial Perception	6.9 ± 2.2	10.7 ± 2.0	−2.539	0.011 *
Praxis	12.9 ± 8.3	25.6 ± 12.3	−2.666	0.008 **
Visual Motor Organization	9.8 ± 2.9	21.3 ± 6.4	−2.668	0.008 **
Thinking Processes	12.6 ± 5.7	24.7 ± 6.7	−2.684	0.007 **
**MVPT-4**				
Visual perception	14.9 ± 5.7	23.0 ± 6.1	−2.675	0.007 **

SD for standard deviation; negative z values (Wilcoxon results) indicate improvements from pre- to post-test; * *p* < 0.05, ** *p* < 0.01.

## Data Availability

Data available on request due to ethical reasons.

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
