# Peer review of "Cognitive Rehabilitation Improves Functional Vision Skills in Children with Cerebral Visual Impairment"

_brainsci, 2025, doi:10.3390/brainsci15060590_

Round 1
Reviewer 1 Report
Comments and Suggestions for Authors
Cognitive Rehabilitation Improves Functional Vision Skills in 2 Children with Cerebral Visual Impairment
General remarks
This paper discussed a preliminary/exploratory study about a cognitive remediation therapy intervention that is aimed at improving functional vision in children with reduced vision, due to CVI. The connection between vision and cognition is one to be attentive to and is promising in improving people’s quality of life. The proposed intervention could contribute in a seemingly efficient way to clinical care for children with CVI. It also contributes to the understanding of the connection between vision and cognition.
The theoretical background and the connection or distinction between vision and cognition needs to be elaborated on and further fundament the hypothesis, conclusions and strengths of the study and the intervention. Also the methods require some more details in order to be fully able to replicate the study. Conclusions are in line with the results. However, the authors need to be more cautious in stating the conclusions, giving their design and small sample size. References are often older than 5 years.
There is an overall merit in this paper to be published, both scientifically and clinically.
Line by line comments
Introduction
Overall, in the introduction, the fact that a large part of the brain is involved in vision / that the brain is high important for seeing well, needs be more elaborately discussed. This would help understanding the overall paper better. The authors should also be more clear on where they think vision ends and cognition starts. Where do you draw the line and how is it defined in your paper?
- Line 35-43
- Elaborate on the role of the brain in vision.
- Line 44-57
- No comments, but overall point about connection between brain, vision and cognition needs to be more clearly explained.
- Line 70-76
- 72-73: be more clear on what visual outcomes. Aka, are they the same as the visual outcomes you use? What do the authors say about the underlying mechanisms of this visual improvement by cognitive improvement?
- Line 77-85
- 77-79: Again you would help the reader understand your line of reasoning by explaining which functional vision outcomes you think that are underlying attention, visual perception and executive functioning. This also implies that there may be functional vision outcomes that may not improve by CRT.
- 79-81: how would this happen? This needs more explanation, as this seems to vague to me.
- 85: Be clear also in your introduction about how you define low vision in this group. You may want to mention this earlier in the introduction that you are writing about children with low vision as a consequence of CVI.
Materials and Methods
- Line 88-99
- 93-94: how as this done?
- Line 100-109
- 101-102: why was this age range chosen?
- 102-103: Was their acuity measured with their current glasses (if they wore glasses), or with optimal refraction?
- 103-104: how was this judged and when in the process was this done?
- 105-107: who performed the assessment regarding cognitive and motor impairments. Was a cut-off used for score indicating ‘severe’ impairments? Where did the cuf-off lie? What does irregular attendance mean exactly?
- 107-109: I would suggest to mention this earlier in the methods. Possibly even between lines 88-99, but at least at the start of this paragraph. Not only because it is chronologically more appropriate, but also because as a reader you are wondering about this, while reading and understanding the inclusion and exclusion criteria.
- Small remark: in the abstract it was mentioned that 9 children were included, in the full text, participant numbers are only discussed in the results. Because I read the abstract first, I was wondering about why that was not mentioned in the methods/participants section.
- Line 110-113
- 111: does ‘diagnosis’ only concern vision of CVI related diagnosis? Please be clear about that. It should be something different than medical history.
- Line 129-130: How are score calculated? It is needed to be able to read the Table 1 in the results section. Moreover, I would advice being more specific about how each of the subdivisions are measured/assessed, or clearly refer to a freely available manual. This would count for all the measures, but is particularly important for your primary outcomes, and for the interpretation of the results. For example, focusing and maintaining focus could easily be seen or interpreted as a purely cognitive skill, which may not necessarily be the case.
- Line 137-144: how is the performance translated to which numerical score? And how are those scores translated into functional skill levels? Are the skill levels what you mention in the beginning of this paragraph (weak-good)?
- Line 145-147: by whom are the children assessed?
- Line 149-158: How are score calculated? It is needed to be able to read the Table 1 in the results section.
- 157-158: This needs a reference, or it needs to be more clear that the paper mentioned in this paragraph also indicated those results.
- Line 160-171: How are score calculated? It is needed to be able to read the Table 1 in the results section.
- 169-171: Like my previous comment, it at least needs to be more clear where you get this information from. It is likely in the paper you already referenced to, but that is not clear enough.
- Line 178-187: who performed the interventions? Where all session done by the same person? What did they know about the study that was being done?
- Line 196-201: given your small sample, it is advisable to add a effect size measure fore each test, if you decide not to, I would add an explanation for that too.
- 199: why use percentages in such a small sample. I might not have overseen it, but I do not see any percentages back in the results section (which is a good thing, because in this sample percentages don’t really have much meaning). I would just remove saying that you calculate percentages.
- Line 202-206: I would begin this heading with this information. Sometime throughout the whole paper, the structure seems a bit off. It is not wrong, but I would advice to critically look at the order of some of the paragraphs, especially in the methods and materials section.
- Line 207-210
- 210: why do you have a reference for SPSS and G*Power (I think you should), but not for ChatGPT. What version did you use, was it a full version, or a free version?
Results
- Line 213-217: Do you have any information about the excluded children? Would it be possible that you a ‘distinctive group remained? For example, younger children, children with different cognitive levels etc. Make sure to discuss this in the discussion, if called for too.
- 213: who makes the referrals and on the basis of what? Were there any children that would have met the criteria, but were not referred?
- 215: can you disclose information about why the child was withdrawn?
- Line 233-241
- 233: Depending on the wishes of the journal editor too, once you have introduced an abbreviation, you should keep on using just the abbreviation, that is not always the case, such as in this sentence.
- Line 256-260: In your methods you mention that you do a Wilcoxon Test on every outcome, due to the structure of the paragraph on analysis I interpret that that also counts for the qualitative measures. However, I do not see any tests about this in the results.
Table 1: misses the explanation of the abbreviation in the table, since this is customary to do. I would also add the N. Was the N the same for all skills? (if not, explain why).
Figure 1: Very insightful figure, makes the results easily visually clear. One question: in the pre test (and post), do I interpret correctly that none of the children score the lowest of the categories (none, weak, moderate, or good)? That is something I would advice to also be explicit about.
Discussion
Overall: Where do you draw the line between (functional) vision and cognition? When is something considered vision and when is something considered something else. Could you really distinguish between vision and visual perception?
Next to that, I would elaborate about the clinical applicability and relevance of your promising intervention. Why would this intervention have preference (it looks like it is easy to implement for example) and what could be possible benefits.
- Line 279-284
- Could you provide some references backing up your statement that there is a growing body of evidence?
- Line 299-312
- 299-301: This needs more explanation: It is not completely clear to me why this is stated. What are the pro’s (and con’s) of focusing on multiple domains? What have been the effect on the results in both your and the cited study?
- 301-305: I am all for performing studies in real life context. However, it could also have some negative consequences. Good to discuss them too, especially considering the low sample size.
- 311-312: could you provide a reference for this?
- Line 313-320: I would advise to be a little more cautious about this direct link with neuroplasticity. From your results you cannot conclude that there really is an effect of the training in the brain, made possible by neuroplasticity.
- Line 321-329: doesn’t this suit the previous paragraph better? It is unclear to me why a new paragraph started. If this first sentence is not related to the previous paragraph, then please make that more clear. In in the next sentence, the ‘moreover’ doesn’t seem to fit the previous sentence well, because you end with the neuroplasticity part and start about quality of life and other plusses of the intervention. Include a short introductory sentence about the plusses of the intervention that could help the reader too with the interpretation here.
- Line 330-344: what about the non-significant outcomes other than writing? Reading receipts and identifying people do seems like totally different skills to me, do you have explanation for the non-significant finds regarding these skills?
- Line 345-349
- 348-349: Could you discuss the possible influence on the results of the external factor that you mention here?
- Are there any more limitations to discuss? Obviously the small sample size for example. Related to that, in the discussion you discuss the significant and non-significant outcomes. Even though you did a power analysis, I would still advise to add some kind of effect size measure (or explain why you did not do that) to put next to the significance of the outcomes. And this discuss this keeping in mind the small sample size. I would also advise a more critical review of the protocol.
- Line 350-369
- 353: maybe mention a possible placebo effect within line 345-349; since a control group may help identifying whether there is a placebo effect.
- 353-355: or to delineate that vision and cognition are very much related and integrated. Studies have shown that completely pulling them apart is very difficult. When vision improves, cognition improves and vice versa. That’s why overall I miss in your paper where you draw that line and how you do that. And this notion is important in understanding what you think is the mechanism behind your intervention. (I would even not call this an limitation (line 356). Possibly also incorporate this in a clinical appraisal of the interventions, since it does not only improve functional vision, but also the cognitive skills.
- 356: I think it is good mention this is in exploratory study earlier on, even in the introduction. It is not clear there it is an exploratory study and you appraise it like that, until the very end of the discussion. Not being clear about this from the beginning, may yield some unrealistic expectations from the reader.
Conclusion
- Line 371-375
- 371: same as with ‘exploratory’ in line 356. You only mention this is an preliminary study in the conclusion. I would do this much earlier, preferably in the introduction. I would also choose one term, either exploratory or preliminary.
Author Response
Dear Reviewer 1,
We would like to sincerely thank all reviewers for their thoughtful and constructive feedback. We greatly appreciate the time and expertise invested in evaluating our manuscript. The comments provided not only acknowledged the strengths of the work but also offered valuable suggestions that have significantly improved the clarity, structure, and depth of the final version.
In the following document, we have addressed each comment point by point. To aid transparency:
- Yellow highlights indicate amendments made in response to editor’s comments.
- Green highlights indicate amendments made in response to Reviewer 1’s comments.
- Turquoise highlights indicate amendments made in response to Reviewer 2’s comments.
- Pink highlights highlights indicate amendments made in response to Reviewer 3’s comments.
- Teal highlights indicate amendments made in response to Reviewer 4’s comments.
We hope that the revised manuscript meets the expectations outlined in the review process, and we remain grateful for your insightful and supportive engagement with our work.

Reviewer 2 Report
Comments and Suggestions for Authors
The paper has the following issues:
1.As a review paper, the number of references is insufficient. Suggest the author to add recent literature, highlight the research status in this field, and increase the timeliness of the paper.
2."In the text, reference numbers should be placed in square brackets [ ] and placed before the punctuation; for example [1], [1–3] or [1,3]. For embedded citations in the text with pagination, use both parentheses and brackets to indicate the reference number and page numbers; for example [5] (p. 10), or [6] (pp. 101–105)"should be deleted.
3.English needs polishing. "Children were eligible for inclusion if they met the following criteria" shoule be "The children were eligible for inclusion if they met the following criteria".
4.The language in section 2.2 is limited and does not express the meaning of "tools". It is suggested that the author further improve it.
5.The conclusion language is too short, without summarizing the research results of the paper or highlighting future research directions. It is recommended that the author further condense and improve it.
Author Response
Dear Reviewer 2,
We would like to sincerely thank all reviewers for their thoughtful and constructive feedback. We greatly appreciate the time and expertise invested in evaluating our manuscript. The comments provided not only acknowledged the strengths of the work but also offered valuable suggestions that have significantly improved the clarity, structure, and depth of the final version.
In the following document, we have addressed each comment point by point. To aid transparency:
- Yellow highlights indicate amendments made in response to editor’s comments.
- Green highlights indicate amendments made in response to Reviewer 1’s comments.
- Turquoise highlights indicate amendments made in response to Reviewer 2’s comments.
- Pink highlights highlights indicate amendments made in response to Reviewer 3’s comments.
- Teal highlights indicate amendments made in response to Reviewer 4’s comments.
We hope that the revised manuscript meets the expectations outlined in the review process, and we remain grateful for your insightful and supportive engagement with our work.

Reviewer 3 Report
Comments and Suggestions for Authors
This paper argues that cognitive rehabilitation therapy (CRT) helps children with cerebral visual impairment (CVI).
20 (as per the summary, 19 as per p 5 line 213) children aged 7-11 were to be investigated, but only 9 of these fulfilled the training in such a way that statistical analysis could be applied. There was no good explanation why these children were chosen and control group.
Looking at the hard-to-read Fig. 1 it is found that for all the measured "Vision skills," at least one child was being "Good" at each of the skills before the training started. If that is the same child in all cases (which ought to be discussed, but is not), that child will not be of significance for the statistical evaluation. Similarly, after the training, at least one child is weak in each skill. If that is the same child for all skills, then the significant contribution from that child would be zero.
How did the children attending ordinary school perform as compared with those who didn't?
Hence, less than 50% of children have been evaluated out of 20, which is a very high failure rate. The authors' conclusion that "functional visual improvement can be improved by training", is, therefore, in my opinion, not valid for all diagnosed children, but only for some. Performing statistical analysis on a sample of 7-9 persons can only be indicative, even if the results are [would be] interesting.
A much more needed discussion on how to select children for successful training is missing. How do we avoid a 50-65% failure rate? Nevertheless, the paper is interesting and points to essential possibilities.
Minor points in order as they appear in the paper:
p3, line 130: "each" is improper. The (two) main dimensions can be further divided into 14...
p4, , line 157: The short notation of the alpha value defined on p. 3 line120 is not used.
p 4 line 160: Section numbering is wrong. 2.2.3
p 5 lines 202-206: Here "Results" are mixed with "Analysis methods". Omit.
p 6 lines 230-231: Some of this text ought to be in the caption of Table 1 for easier reading.
The criterion for the z-value is not explained. What does the value mean? Help the reader!
line 240: Maybe it takes quite some time to develop better writing, once the vision is improved? Discussion needed.
line 230 and Table 1: line 230 says "standard deviation" and Table 1 says Sum of Squares (SS). These concepts certainly are related but by means the same thing.
Table 1:Numerical accuracy. The numerical accuracy reported is too high. It is no coincidence that the figures after the decimal points for the mean values always are multiples of 11. It is left to the authors to figure out why!
p8 lines 325-329. The discussion on how much personell resources (qualifications and man-hours) which is needed is lacking.
p9 line 322: "outdated" might be "politically correct", but from a scientific point of view the authors have neither shown this to be the case, nor given a valid reference. The authors might want to rewrite lines 322 and 323 in a more nuanced and elegant way.
p9 around line 337. Here again, the time aspect might be discussed.
p 9 line 374: In the present writing, the "a" in ".....is a useful approach...." is missing.
Author Response
Dear Reviewer 3,
We would like to sincerely thank all reviewers for their thoughtful and constructive feedback. We greatly appreciate the time and expertise invested in evaluating our manuscript. The comments provided not only acknowledged the strengths of the work but also offered valuable suggestions that have significantly improved the clarity, structure, and depth of the final version.
In the following document, we have addressed each comment point by point. To aid transparency:
- Yellow highlights indicate amendments made in response to editor’s comments.
- Green highlights indicate amendments made in response to Reviewer 1’s comments.
- Turquoise highlights indicate amendments made in response to Reviewer 2’s comments.
- Pink highlights highlights indicate amendments made in response to Reviewer 3’s comments.
- Teal highlights indicate amendments made in response to Reviewer 4’s comments.
We hope that the revised manuscript meets the expectations outlined in the review process, and we remain grateful for your insightful and supportive engagement with our work.

Reviewer 4 Report
Comments and Suggestions for Authors
The work is devoted to the restoration of cognitive and perceptual functions in children with impaired central visual processes. To improve the manuscript's rating, I recommend paying attention to the following:
- A total of 9 patients. A detailed description should be given in the table for each (without name) - age, anamnesis, more precise diagnosis, other diseases, school grade, etc.
- How was the children's consent obtained? From their parents?
- Figure 1 - low-quality signatures inside.
- The results chapter does not fully disclose the generally interesting work. I do not see the point in developing this in the appendix. This disrupts the perception of the manuscript. Move figure A1 and table A1 to results with a description. Other information from the appendix - either move to methods, or, possibly, delete.
- Conclusion. Specify, show the most important results.
- Does source 30 contain names in Cyrillic?
Author Response
Dear Reviewer 4, We would like to sincerely thank all reviewers for their thoughtful and constructive feedback. We greatly appreciate the time and expertise invested in evaluating our manuscript. The comments provided not only acknowledged the strengths of the work but also offered valuable suggestions that have significantly improved the clarity, structure, and depth of the final version.
In the following document, we have addressed each comment point by point. To aid transparency:
- Yellow highlights indicate amendments made in response to editor’s comments.
- Green highlights indicate amendments made in response to Reviewer 1’s comments.
- Turquoise highlights indicate amendments made in response to Reviewer 2’s comments.
- Pink highlights highlights indicate amendments made in response to Reviewer 3’s comments.
- Teal highlights indicate amendments made in response to Reviewer 4’s comments.
We hope that the revised manuscript meets the expectations outlined in the review process, and we remain grateful for your insightful and supportive engagement with our work.

Round 2
Reviewer 2 Report
Comments and Suggestions for Authors
Accept in present form.
Reviewer 3 Report
Comments and Suggestions for Authors
I checked the authors' answers and quickly the paper. I think it is an interesting paper, despite the low number of participants and being an "exploratory pilot" study.
Reviewer 4 Report
Comments and Suggestions for Authors
The paper looks much better after reviewing and can be recommended for printing.